# Fe_3_O_4_@COF(TAPT–DHTA) Nanocomposites as Magnetic Solid-Phase Extraction Adsorbents for Simultaneous Determination of 9 Mycotoxins in Fruits by UHPLC–MS/MS

**DOI:** 10.3390/toxins15020117

**Published:** 2023-02-01

**Authors:** Jie Wang, Qingwen Huang, Wenbo Guo, Dakai Guo, Zheng Han, Dongxia Nie

**Affiliations:** 1School of Health Science and Engineering, University of Shanghai for Science and Technology, Shanghai 200093, China; 2Institute for Agro-Food Standards and Testing Technology, Shanghai Academy of Agricultural Sciences, Shanghai 201403, China

**Keywords:** covalent organic framework, magnetic solid-phase extraction, mycotoxins, UHPLC–MS/MS, fruits

## Abstract

In this study, a simple and efficient magnetic solid-phase extraction (MSPE) strategy was developed to simultaneously purify and enrich nine mycotoxins in fruits, with the magnetic covalent organic framework nanomaterial Fe_3_O_4_@COF(TAPT–DHTA) as an adsorbent. The Fe_3_O_4_@COF(TAPT–DHTA) was prepared by a simple template precipitation polymerization method, using Fe_3_O_4_ as magnetic core, and 1,3,5-tris-(4-aminophenyl) triazine (TAPT) and 2,5-dihydroxy terephthalaldehyde (DHTA) as two building units. Fe_3_O_4_@COF(TAPT–DHTA) could effectively capture the targeted mycotoxins by virtue of its abundant hydroxyl groups and aromatic rings. Several key parameters affecting the performance of the MSPE method were studied, including the adsorption solution, adsorption time, elution solvent, volume and time, and the amount of Fe_3_O_4_@COF(TAPT–DHTA) nanomaterial. Under optimized MSPE conditions, followed by analysis with UHPLC–MS/MS, a wide linear range (0.05–200 μg kg^−1^), low limits of detection (0.01–0.5 μg kg^−1^) and satisfactory recovery (74.25–111.75%) were achieved for the nine targeted mycotoxins. The established method was further successfully validated in different kinds of fruit samples.

## 1. Introduction

Mycotoxins are toxic secondary metabolites produced by toxigenic fungi under suitable environmental conditions. Fruits increasingly favored by consumers, owing to their high moisture content, rich nutrition and improper harvest or storage conditions, are highly susceptible to various fungi such as *Alternaria, Aspergillus, Fusarium*, etc. [1,2]. Among these metabolites, aflatoxin B_1_(AFB_1_), ochratoxin A (OTA), zearalenone (ZEN) and *Alternaria* toxins, mainly including tentoxin (TEN), altenuene (ALT), altenusin (ALS), alternariol monomethyl ether (AME), alternariol (AOH) and tenuazonic acid (TeA), are the most frequently found in various fruits [3,4]. Those mycotoxins can cause acute and chronic toxic effects (teratogenicity, cytotoxicity, reproductive and developmental toxicity, etc) on animals and human [5,6]. Owing to mycotoxins’ high toxicity and widespread contamination of food, the European Food Safety Authority (EFSA) has established a threshold of toxicological concern (TTC) values of 2.5 ng/kg bw/day for AOH and AME, and 1.5 μg/kg bw/day for TeA [7]. Maximum levels of AFB_1_, OTA and ZEN in different types of food have been set in the European Union, USA, Canada, China and other countries [8,9]. Given the widely occurring co-contamination of mycotoxins in various fruits, it is imperative to establish an efficient analytical method to simultaneously determine multiple mycotoxins in different kinds of fruits, such as watermelon, hawthorn, melon, tomato, strawberry, etc. [10,11].

Conventional methods for the analysis of mycotoxins in foodstuffs commonly rely on high-performance liquid chromatography (HPLC) coupled with ultraviolet, fluorescence or mass spectrum detectors [12,13]. Nevertheless, the matrix of fruit samples contains large amounts of pigments, cellulose and minerals, which could dramatically impede the detection of trace mycotoxins in food. Therefore, efficient enrichment and purification of multiple mycotoxins in fruit samples is crucial before instrumental analysis. Recently, the most commonly used sample pretreatment methods have included liquid-liquid extraction (LLE) [14], solid-phase extraction (SPE) [15] and QuEChERS methods [16]. Among these methods, immunoaffinity SPE approach has high specificity, but it is also expensive, time consuming and, in particular, not applicable to the simultaneous purification of multiple mycotoxins. Magnetic solid phase extraction (MSPE), as a new kind of SPE, has attracted great attention by virtue of its easy separation, convenient operation and time-saving qualities [17]. In the MSPE process, the magnetic sorbents are directly dispersed in the sample solution for rapid and efficient extraction of analytes, and then quickly separated by an external magnetic field. Many kinds of magnetic nanomaterials have already been developed to enrich targeted analytes and eliminate matrix interferences [18,19,20].

Covalent organic frameworks (COFs) are a new class of crystalline material, which can be constructed with organic building units by covalent bonds of elements (C, O, N, H, etc.) [21]. COFs, which have been shown to exhibit many unique characteristics such as large specific surface areas, permanent porosity, rich functional groups, and good thermal and chemical stability, have received increasing attention in the field of sample pretreatment [22,23]. Given that the structure and surface properties of COFs has been mainly dependent on covalent linkage topology schemes and organic monomers, considerable attention has been focused on the exploration of various synthetic strategies in recent years [24,25]. For instance, Xu et al. demonstrated the superiority of triazine-based COF for the extraction of phenoxy carboxylic acid pesticide residue [26]. Li and co-workers synthesized Fe_3_O_4_@COF(TpDA) material to enrich plant growth regulators from fruits and vegetables through π-π and hydrogen bonding interactions [27]. Nevertheless, owing to the low concentrations, different functional groups and polarity of various mycotoxins, it is still challenging to achieve the simultaneous and efficient extraction of multiple mycotoxins from complex matrices. To date, the COF-based MSPE methods for the determinations of mycotoxins have rarely been studied [28].

In this study, using Fe_3_O_4_ as magnetic core, and 1,3,5-tris-(4-aminophenyl) triazine (TAPT) and 2,5-dihydroxy terephthalaldehyde (DHTA) as two building units, Fe_3_O_4_@COF(TAPT–DHTA) was firstly designed and applied as an MSPE adsorbent to extract nine mycotoxins (Appendix A) from fruits. The targeted mycotoxins were then analyzed by UHPLC–MS/MS (Appendix A). The prepared Fe_3_O_4_@COF(TAPT–DHTA) adsorbents were expected to effectively enrich mycotoxins by virtue of their abundant hydroxyl groups and aromatic rings. The features of Fe_3_O_4_@COF(TAPT–DHTA) were characterized and several key factors affecting MSPE were optimized. In addition, the possible adsorption mechanism was also discussed. The proposed method was further validated and applied to the analysis of mycotoxins in fruits including watermelon, hawthorn, melon, tomato and strawberry. The schematic of the fabrication and application of Fe_3_O_4_@COF(TAPT–DHTA) is shown in Figure 1.

## 2. Results and Discussion

### 2.1. Characterization of Fe_3_O_4_@COF(TAPT–DHTA)

The morphology of the prepared nanomaterial was characterized by SEM (Figure 2A,B). It can be seen that Fe_3_O_4_ has a regular spherical structure with a diameter of approximately 200 nm. However, Fe_3_O_4_@COF(TAPT–DHTA) exhibited a dense surface with a significantly larger particle size of nearly 850 nm, proving that COFs were successfully grafted onto Fe_3_O_4_ nanoparticles.

The formation of Fe_3_O_4_@COF(TAPT–DHTA) was verified by FT-IR spectroscopy (Figure 2C). For Fe_3_O_4_, the characteristics peak of the Fe–O–Fe vibration was observed at 588 cm^−1^. For Fe_3_O_4_@COF(TAPT–DHTA), the peaks at 1670 cm^−1^ and 3329 cm^−1^ were assigned to the C=O stretching vibration of DHTA and N–H stretching of TAPT, respectively [26]. The additional typical peaks appeared at 1367 cm^−1^ and 1501 cm^−1^ owing to the presence of the triazine ring [29]. Meanwhile, the stretching bands at 1620 cm^−1^ might be ascribed to the formation of imine bonds, proving the successful condensation of formyl linker and amine node [26,29].

Figure 2D shows the UV–VIS absorption spectra of Fe_3_O_4_ and Fe_3_O_4_@COF(TAPT–DHTA), respectively. Compared with Fe_3_O_4_, Fe_3_O_4_@COF(TAPT–DHTA) showed an absorption peak at 345 nm, which was attributed to the existence of the conjugated double bond in COF. The strong conjugated double bond made Fe_3_O_4_@COF(TAPT–DHTA) nanocomposites ideal adsorbents for the extraction of benzene ring-containing mycotoxins. In fact, after capturing nine mycotoxins (Appendix A), the absorption peak of Fe_3_O_4_@COF(TAPT–DHTA) at 345 nm was red-shifted to 351–358 nm, indicating the strong π-π-stacking interaction between the nanocomposites and targeted mycotoxins [30].

Furthermore, the energy dispersive X-ray spectroscopy (EDAX) line-scanning method was conducted to analyze the elements contained in Fe_3_O_4_@COF(TAPT–DHTA) (Figure 2E). The results demonstrated that the prepared nanomaterial contains four elements, Fe, O, C and N, which also proved the successful synthesis of the nanomaterial. 

The magnetic properties of Fe_3_O_4_ and Fe_3_O_4_@COF(TAPT–DHTA) were also investigated, and the magnetization curves are shown in Figure 2F. The saturation magnetization values of Fe_3_O_4_ and Fe_3_O_4_@COF(TAPT–DHTA) were 46.542 and 18.457 emu g^−1^, respectively. Although the magnetization values of Fe_3_O_4_@COF(TAPT–DHTA) decreased compared with pure particles, it could still provide sufficient magnetism for MSPE. As shown in the inset of Figure 2F, Fe_3_O_4_@COF(TAPT–DHTA) was well-dispersed in the solvent, and could be collected within 30 s by an external magnetic field.

### 2.2. Optimization of MSPE Conditions

Fe_3_O_4_@COF(TAPT–DHTA) nanomaterial, with rich hydroxyl groups and aromatic rings, was utilized as competent adsorbent in MSPE for the selective enrichment of multiple mycotoxins. To achieve excellent MSPE procedure, several important parameters were studied by using spiked tomato samples (20 μg kg^−1^), including adsorption solution, pH, ionic strength, adsorption time, elution solvent, elution volume and time, and Fe_3_O_4_@COF(TAPT–DHTA) amount.

#### 2.2.1. MSPE Adsorption Solution

The contents of organic solvent in the adsorption solution played a crucial role in the adsorption process. Therefore, the effect of different contents of acetonitrile (0%, 1%, 2%, 3%, 4% and 5%, *v*/*v*) were examined. As shown in Figure 3A, when 1% acetonitrile in water was applied, the highest recoveries of targeted mycotoxins except for AFB_1_ were obtained (83–104%). For AFB_1_, there was no significant difference in recovery between 1% acetonitrile (79%) and 2% acetonitrile (81%) (*p* > 0.05, Appendix A). To ensure adsorption efficiency of the Fe_3_O_4_@COF(TAPT–DHTA) composite for all the targeted analytes, 1% acetonitrile in water was used as the adsorption solution.

The pH value of the sample solution had a great influence on the charge property of the surface of the Fe_3_O_4_@COF(TAPT–DHTA), as well as on the stability and existing forms of the targeted mycotoxins. Figure 3B shows the recoveries of the nine mycotoxins in different pH values. The recoveries first increased and then decreased when the pH varied from acidic to basic. When pH was set as 4.0, satisfactory recoveries (76–104%) were achieved for all nine mycotoxins. Although the recoveries for ALS and TEN obtained at pH 5.0 were a little higher than those at pH 4.0, no significant difference was found for them (*p* > 0.05, Appendix A). This result might be due to the characteristics of analytes and nanomaterial [26]. As can be seen in Appendix A, the mycotoxins had polar hydroxyl groups and carboxyl groups with pKa values between 3.08 and 7.58, which were not conducive to the formation of hydrogen bonds with Fe_3_O_4_@COF (TAPT–DHTA) under alkaline conditions. Therefore, the optimal pH was set to 4.0.

To explore the effect of ionic strength on the extraction efficiency of mycotoxins, different sample solutions with various amounts of sodium chloride (0, 2, 4, 6, 8, and 10 mg mL^−1^) were evaluated (Figure 3C). It was observed that the recoveries of the targeted mycotoxins significantly decreased with the increased salt concentration. Only in the case of no salt addition were the highest recoveries (76–103%) obtained for all nine mycotoxins. In particular, the recoveries for six mycotoxins (TeA, TEN, ALT, AFB_1_, OTA and ZEN) were much higher than those obtained when the adsorption solution contained sodium chloride (*p* < 0.05, Appendix A). This phenomenon might be explained by the fact that the intermolecular aggregation of mycotoxins was enhanced owing to the addition of salt ions, which inhibited the further adsorption of isolated mycotoxin molecules onto the Fe_3_O_4_@COF(TAPT–DHTA) [28]. Hence, NaCl was not used for subsequent experiments.

The adsorption time could also affect the extraction efficiency. A series of experiments was conducted to optimize the adsorption times (2, 4, 6, 8 and 10 min). It was observed that the recoveries of the nine mycotoxins gradually increased in the range of 2–8 min (Figure 3D). When 8 min was used, satisfactory recoveries (78–96%) were obtained for all mycotoxins. When the time was further prolonged, no significant improvement could be observed (*p* > 0.05, Appendix A). Therefore, 8 min was finally determined for further experiments.

#### 2.2.2. The Amount of Fe_3_O_4_@COF(TAPT–DHTA)

The amount of adsorbent is a critical factor in the MSPE process. The effects of various amounts (10, 15, 20, 25, 30 mg) of Fe_3_O_4_@COF(TAPT–DHTA) nanomaterial on the recoveries of the nine mycotoxins were compared (Figure 3E). When 20 mg Fe_3_O_4_@COF(TAPT–DHTA) was used, the adsorption and desorption of the nine mycotoxins achieved equilibrium, and the highest recoveries were achieved within the acceptable range (75–97%) except for AFB_1_. Although the recovery of AFB_1_ was a little lower than that in the case of 30 mg, there was no significant difference between them (*p* > 0.05, Appendix A). When the amounts were 10 and 15 mg, Fe_3_O_4_@COF(TAPT–DHTA) could not provide adequate adsorption sites for these mycotoxins. Conversely, when the amounts of adsorbent were excessive, such as 25 and 30 mg, it might not have been easy to elute these mycotoxins from Fe_3_O_4_@COF(TAPT–DHTA). Thus, 20 mg of Fe_3_O_4_@COF(TAPT–DHTA) was selected for study.

#### 2.2.3. Elution Solvent and Time

Depending on the effect of the adsorption mechanism of the targeted mycotoxins on the Fe_3_O_4_@COF(TAPT–DHTA) nanomaterial, the π-π, hydrogen bonding and hydrophobic interactions between adsorbent and analytes could be destroyed by organic solvents. Therefore, elution solvent with different polarities was optimized. As indicated in Appendix A, when methanol/acetonitrile/formic acid (80/19/1) was used, the recoveries of seven mycotoxins (TeA, AOH, AME, TEN, ALT, ALS and OTA) were acceptable in the range of 83–107%. However, the recoveries of AFB_1_ and ZEN were much lower compared to methanol (*p* < 0.05, Appendix A). Only pure methanol used as elution solvent could achieve satisfactory recoveries for all nine of the mycotoxins (81–99%). The best elution performance of methanol is attributed to the strong polarity, which could compete for the hydrogen binding sites with mycotoxins and lead to the destruction of the formed hydrogen bonding between adsorbent and mycotoxins.

The volume of the elution solvent is crucial in the MSPE procedure. Different volumes (3, 9, 15 mL) of elution solvent were investigated (Appendix A). The results demonstrated that 3 mL elution solvent was enough to achieve satisfactory recoveries (78–99%). With the increase of the elution volume (9 and 15 mL), the recoveries of the targeted mycotoxins were not obviously improved (*p* > 0.05, Appendix A). In addition, different elution time (1, 2, 3, 4 and 5 min) was studied, as shown in Figure 3F. The recoveries of these mycotoxins increased with an increase of elution time ranging from 1 to 4 min, and then remained stable (*p* > 0.05, Appendix A), suggesting the complete elution of the targeted mycotoxins from Fe_3_O_4_@COF(TAPT–DHTA) nanomaterial. Consequently, 3.0 mL of pure methanol for 4 min were the optimal elution conditions.

Therefore, based on the optimization of these key parameters, the optimal MSPE conditions were determined as follows: 20 mg Fe_3_O_4_@COF(TAPT–DHTA) as magnetic adsorbent, extraction time of 8 min using 3 mL water containing 1% acetonitrile as adsorption solution (pH 4.0), and 3 mL methanol as elution solvent with desorption time of 4 min.

### 2.3. Method Validation

For further characterization of the MSPE performance of Fe_3_O_4_@COF(TAPT–DHTA), the purification efficiency of the synthesized nanomaterials was investigated in five kinds of fruits including watermelon, hawthorn, melon, tomato and strawberry. The sample solutions treated with Fe_3_O_4_@COF(TAPT–DHTA) were almost colorless and transparent (Figure 4A) in comparison with the untreated solutions, indicating that Fe_3_O_4_@COF(TAPT–DHTA) could eliminate the interferences. The matrix effects of the targeted mycotoxins in the five kinds of fruits were in the range of 80.04–106.87% except for TeA (127.19–144.88%), ALS (46.31–102.74%) and ZEN (55.01–83.33%), as shown in Figure 4B. To achieve accurate quantification, matrix-matched calibration curves were applied to compensate for the matrix effects in this study.

The current analytical method demonstrated an optimal selectivity, because there were no apparent interfering peaks presenting near the retention time of all the targeted mycotoxins in the tomato matrix (Appendix A). Excellent linearity was obtained for all the analytes, with correlation coefficients (R^2^) > 0.99 over the concentration range of 0.05–200 μg kg^−1^ in the matrices for all five kinds of fruits (Table 1). The LODs of the nine mycotoxins ranged from 0.01 to 0.5 μg kg^−1^, and the LOQs ranged from 0.05 to 1.0 μg kg^−1^. Acceptable recoveries in the range of 74.25–111.75% were also obtained. The intra- and inter-day precisions were in the range of 2.08–9.01% and 2.22–12.92%, respectively (Table 2). All the above results indicated that the established UHPLC–MS/MS method had high sensitivity, accuracy and precision, and could be applied for the simultaneous quantitative analysis of multiple mycotoxins in different fruits. Compared with previously reported methods, the proposed method possesses equal or even greater sensitivities and recoveries (Appendix A).

### 2.4. Method Application

A validated approach was employed to investigate mycotoxin contamination of a total of 100 samples of the five kinds of fruits, i.e., tomato, strawberry, watermelon, melon and hawthorn. As shown in Appendix A, ALT was the most frequently detected mycotoxin with incidences (concentration ranges) of 40% (2.2–44.5 μg kg^−1^), 55% (3.4–54.8 μg kg^−1^), 25% (29.5–56.3 μg kg^−1^), 50% (43.4–123.7 μg kg^−1^) and 45% (38.8–190.4 μg kg^−1^) in the tomato, strawberry, watermelon, melon and hawthorn samples, respectively. TeA was detected in the tomato and strawberry samples, with incidences (concentration ranges) of 50% (3.8–6.5 μg kg^−1^) and 45% (1.9–5.6 μg kg^−1^), respectively. The incidence (concentration ranges) of AOH in the tomato, strawberry and hawthorn samples was 10% (3.05–4.0 μg kg^−1^), 15% (4.9–20.0 μg kg^−1^) and 25% (3.7–14.2 μg kg^−1^), respectively. These results are similar to previous reports, with pollution levels at the same level as other regions [31,32]. In addition, concentrations of AME, TEN and ALS were detected in the range of 1.4–16.8 μg kg^−1^, 0.6–18.2 μg kg^−1^ and 1.3–8.1 μg kg^−1^, respectively. OTA, AFB_1_ and ZEN were not detected in the five kinds of fruit samples. Therefore, the survey results demonstrated that the fruits were mainly susceptible to contamination by *Alternaria* mycotoxins, which might impose health risks to the consumer.

To demonstrate the accuracy of the method, a comparison between the current established approach and the reference method [33] was performed on typical tomato samples (nos. 7, 14 and 20). There was no significant difference between the results obtained by the two methods, with the RSDs lower than 10% (*p* > 0.05, Appendix A), verifying the accuracy and applicability of the developed UHPLC–MS/MS method.

## 3. Conclusions

In summary, Fe_3_O_4_@COF(TAPT–DHTA) was synthesized and utilized as an MSPE adsorbent for the first time to enrich and determine trace levels of nine mycotoxins in fruits by coupling with UHPLC–MS/MS. Owing to the presence of abundant aromatic rings and the carbonyl group in the structure of the adsorbent, the effective enrichment of the targeted mycotoxins was achieved by virtue of a strong π-π interaction and hydrogen bonding between the mycotoxins and Fe_3_O_4_@COF(TAPT–DHTA). The established method showed a wide linear range, high sensitivity, satisfactory recoveries and good precision, and was successfully employed for the analysis of mycotoxins in real fruit samples. The fruits were found to be easily contaminated with different mycotoxins, i.e., TeA, AOH, ALT, etc. Therefore, continuous monitoring of the occurrence of multiple mycotoxins is essential to ensure of the safe consumption of fruits.

## 4. Materials and Methods

### 4.1. Chemicals and Materials

All the organic solvents, acids, alkalis and salts were HPLC or analytical grade. Acetonitrile and methanol were purchased from Merck (Darmstadt, Germany). Ammonium acetate, formic acid, aqueous ammonia, FeCl_3_·6H_2_O (≥99.5%) and FeCl_2_·4H_2_O (≥99%) were provided by Sinopharm Chemical Reagent Co. (Shanghai, China).1,2-dichlorobenzene, 1-Butanol and 1,4-Dioxane were supplied by Macklin Co. (Shanghai, China). DHTA and TAPT were obtained from Yuanye Bio-Technology Co. (Shanghai, China). Nylon filters (0.22 μm) were obtained from Navigator Lab Instrument Co. Ltd. (New York, NY, USA). Deionized water was prepared by a Milli-Q water purification system (Millipore, Bedford, MA, USA).

High purity (≥98%) standards of TeA, AOH, AME, TEN, ALT, ALS, AFB_1_, OTA and ZEN were obtained from Sigma-Aldrich (St. Louis, MO, USA). Standard stock solutions of the nine mycotoxins (10 μg mL^−1^) were prepared in acetonitrile and stored at −20 °C in the dark. Their chemical structures and physicochemical parameters are shown in Appendix A.

A total of 100 random fruit samples were provided by local markets and supermarkets in Shanghai. The samples were ground into powder or pulp and stored at −20 °C. 

### 4.2. Apparatus and Characterization

Scanning electron microscopy (SEM) images were achieved on a ZEISS Gemini SEM 300 electron microscope (Oberkochen, Baden-Warburg, Germany) operated at 3.0 kV. Fourier transform infrared (FTIR) spectroscopy with a recording range of 500–4000 cm^−1^ was performed with an FTIR spectrometer (FTIR, Thermo Scientific Nicolet iS20, Waltham, MA, USA). Ultraviolet-visible (UV–VIS) absorption spectra were obtained using a JENA2010 spectrophotometer (JENA, Turingia, Germany). A Super-X Spectrometer (Waltham, MA, USA) used for elemental analysis and energy spectrum analysis. The magnetic property was studied with a LakeShore7404 vibrating sample magnetometer (MI, USA).

### 4.3. Preparation of Fe_3_O_4_@COF(TAPT–DHTA)

The preparation procedures of Fe_3_O_4_@COF(TAPT–DHTA) are shown in Figure 1. Firstly, Fe_3_O_4_ magnetic nanoparticles (MNPs) were fabricated according to the previously reported Massart method with minor changes [34,35]. Briefly, 2.6 g FeCl_3_·6H_2_O and 1.59 g FeCl_2_ 4H_2_O were dissolved in 12.5 mL water containing 0.43 mL of 30% HCl under nitrogen atmosphere. Afterwards, the obtained mixture was added dropwise into 125 mL 1.5 M NaOH aqueous solution under vigorous stirring for 40 min. The synthesized Fe_3_O_4_ product was collected using a strong magnet and washed several times with water, then dried at 65 °C for further use. Secondly, Fe_3_O_4_@COF(TAPT–DHTA) was fabricated using a simple synthesis method [36]. The prepared Fe_3_O_4_ MNPs (200 mg), DHTA (105 mg) and TAPT (85 mg) were mixed in the 40 mL 1,4-dioxane/butanol (1/1, *v*/*v*) solvent system, and then sonicated for 5 min. This was followed by adding 0.5 mL of 36% acetic acid, and the mixture was then stirred for 2 h at 25 °C. Then, 4.5 mL acetic acid/deionized water (2/1, *v*/*v*) was dropped into the mixture and refluxed at 75 °C for 48 h. Finally, the obtained Fe_3_O_4_@COF(TAPT–DHTA) was separated by magnet, washed several times with methanol and dried for further use.

### 4.4. Sample Preparation

A fruit sample of 2.0 g was vortexed in 10 mL acetonitrile/formic acid (99/1, *v*/*v*) for 1 min, and then ultrasonicated for 60 min at room temperature (25 °C). After centrifugation at 8000 rpm for 5 min, a 3 mL aliquot of the supernatant was evaporated to dryness under nitrogen flow and dissolved to 3 mL with an aqueous solution containing 1% acetonitrile (pH 4.0). This was followed by the addition of 20 mg of Fe_3_O_4_@COF(TAPT–DHTA), the mixture was vortexed for 8 min to fully adsorb the mycotoxins onto the nanocomposites. Afterwards, the supernatant was discarded by a magnet and 3 mL methanol was added to elute the targeted mycotoxins for 4 min under ultrasonic conditions. The elution was collected and dried with nitrogen at 50 °C. Finally, the residues were re-dissolved with 1 mL acetonitrile/water containing 5 mmol L^−1^ ammonium acetate (20/80, *v*/*v*), passed through a 0.22 μm nylon filter before analysis. The schematic illustration of MSPE is shown in Figure 1. To avoid introducing other impurities, Fe_3_O_4_@COF(TAPT–DHTA) nanocomposites were used as disposable purification adsorbents.

### 4.5. UHPLC–MS/MS Analysis

Chromatographic analysis of the nine mycotoxins was accomplished by an Acquity UHPLC (Waters, Milford, MA, USA) with an analytical column EC-C18 (100 mm × 3.0 mm, i.d. 2.7 μm) (Agilent, Santa Clara, CA, USA) maintained at 40 °C. The binary gradient mixture consisted of (A) methanol and (B) water containing 5 mmol L^−1^ ammonium acetate used as a mobile phase, with a flow rate of 0.4 mL min^−1^. The gradient elution program was set as follows: 0–5 min, 70% B; 5–7 min, 10% B; 7.5–8.5 min, 70% B. The injection volume was 3.0 μL.

Waters XEVO TQ-S mass spectrometer system (Waters, Milford, MA, USA) was performed to detect separated mycotoxins both in positive (ESI^+^) and negative (ESI^−^) mode. The capillary voltages were set at 2.5 kV (ESI^+^) and 1.5 kV (ESI^−^). The source and desolvation temperatures were 150 °C and 500 °C, respectively. The desolvation gas flow and cone gas flow were 800 L h^−1^ and 150 L h^−1^, respectively. Multiple reaction monitoring (MRM) mode was established and is shown in Appendix A, and data analysis was performed using MassLynx v4.1 and Targetlynx (Waters).

### 4.6. Method Validation

The performance of the established method was carefully validated according to the recommendations of European Commission Decision 2002/657/EC by determination of linearity, matrix effect, sensitivity, accuracy (recovery), and precision (%RSD) [37]. Different concentrations (0.5, 1, 2, 5, 10, 20, 50, 100 and 200 ng mL^−1^) of mixed standard solutions of the nine mycotoxins were freshly prepared both in blank matrix solution and standard solutions (acetonitrile/water containing 5 mmol L^−1^ ammonium acetate, 20/80, *v*/*v*), respectively. The calibration curves were obtained by plotting the responses (y) versus analyte concentrations (x). The limit of detection (LOD) and limit of quantification (LOQ) were applied to evaluate the sensitivity of the method, which were defined by the signal-to-noise ratio (S/N) of 3 and 10, respectively. The recoveries and intra- and inter-day precision were measured using blank samples spiked with three different concentrations of mycotoxins (2, 50, and 100 µg kg^−1^). The relative standard deviations (RSDs) in a single day and five consecutive days were devoted to estimation of the intra-day precision and inter-day precision, respectively. The matrix effect (ME) (%) was calculated according to the following equation:ME (%) = (Slope matrix spiked-Slope standard solution)/Slope standard solution × 100%
where Slope matrix spiked and Slope standard solution represented the slope of the calibration curve in the matrix and reagent solution, respectively.

### 4.7. Statistical Analysis

Data were analyzed using SPSS Statistics 18 (SPSS, Inc., Chicago, IL, USA) and presented as mean ± SD. Significant tests were conducted by the analysis of variance (ANOVA) followed by Duncan’s least significant test. The Shapiro–Wilk test was used to check the normality of the data before using ANOVA.

## Figures and Tables

**Figure 1 toxins-15-00117-f001:**
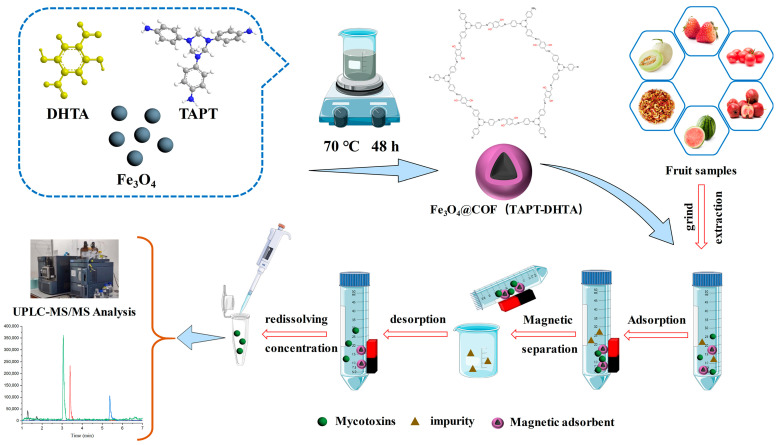
Schematic illustration of the fabrication of Fe_3_O_4_@COF(TAPT–DHTA) and the established MSPE procedure.

**Figure 2 toxins-15-00117-f002:**
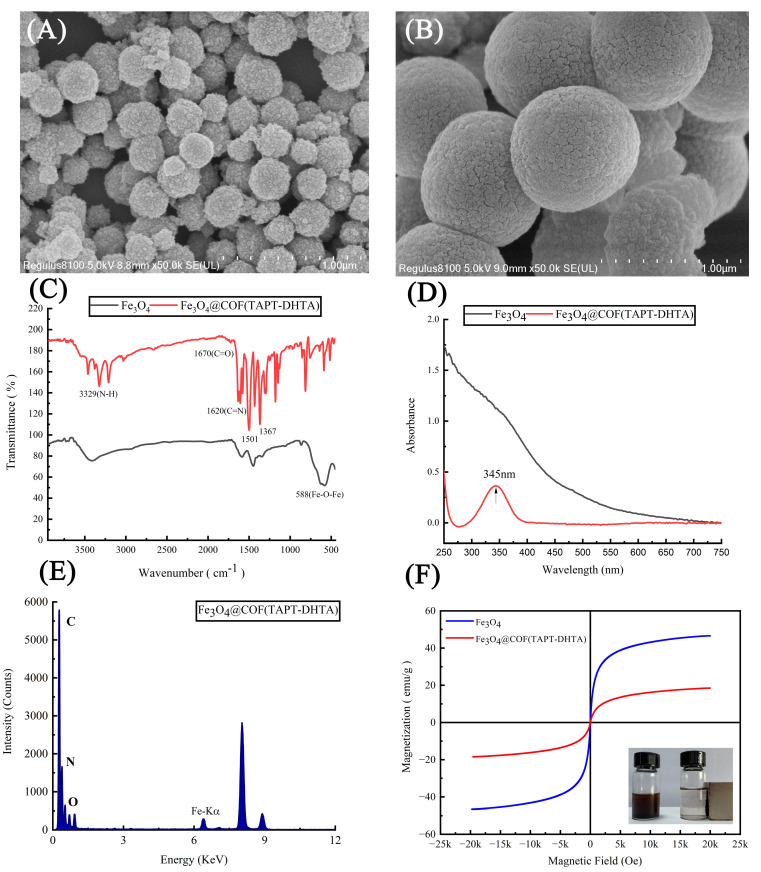
SEM images of Fe_3_O_4_ (**A**) and Fe_3_O_4_@COF(TAPT–DHTA) (**B**); FT-IR (**C**) and UV–VIS (**D**) spectra of the Fe_3_O_4_ and Fe_3_O_4_@COF(TAPT–DHTA); EDAX spectra of Fe_3_O_4_@COF(TAPT–DHTA) (**E**); and hysteresis loops of the Fe_3_O_4_ and Fe_3_O_4_@COF(TAPT–DHTA) (**F**).

**Figure 3 toxins-15-00117-f003:**
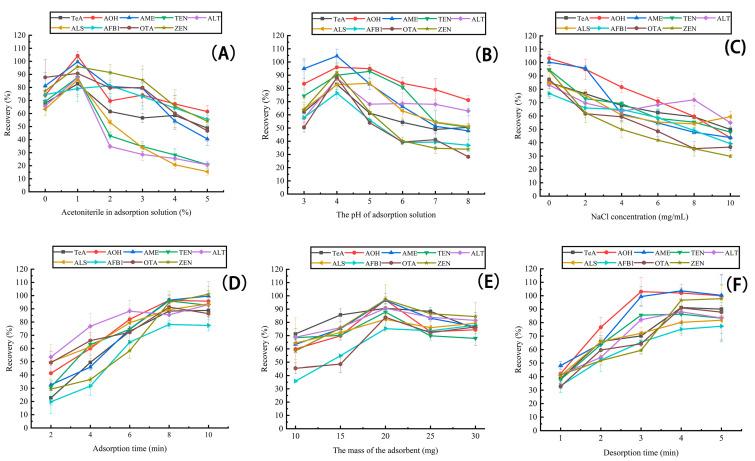
Effects of the key parameters on the performance of Fe_3_O_4_@COF(TAPT–DHTA) MSPE procedure, including (**A**) percentage of acetonitrile in adsorption solution, (**B**) the pH of adsorption solution, (**C**) the concentration of NaCl in adsorption solution, (**D**) adsorption time, (**E**) the amount of adsorbent and (**F**) desorption time. The concentrations of mycotoxins tested were 20 μg kg^−1^ (*n* = 3).

**Figure 4 toxins-15-00117-f004:**
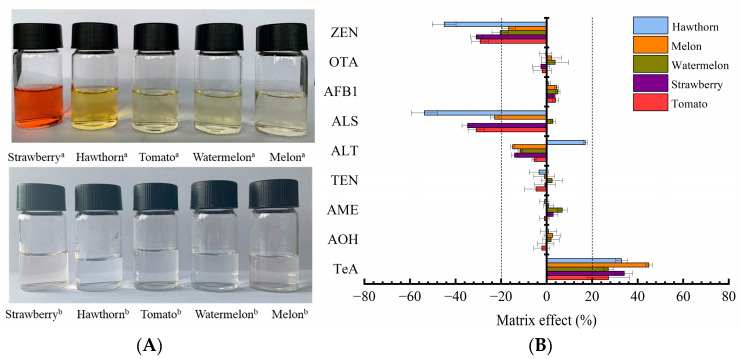
Visual appearance (**A**) and matrix effects (**B**) of 9 mycotoxins in hawthorn, melon, watermelon, strawberry and tomato matrices treated with Fe_3_O_4_@COF(TAPT–DHTA), a without treatment by Fe_3_O_4_@COF(TAPT–DHTA) and b treated with Fe_3_O_4_@COF (TAPT–DHTA).

**Table 1 toxins-15-00117-t001:** Linear range, detection limit and quantification limit of MSPE combined with UHPLC–MS/MS in different matrices.

Matrix	Mycotoxin	Formula	Linear Range (μg kg^−1)^	LOD ^a^ (μg kg^−1^)	LOQ ^b^ (μg kg^−1^)	R^2^
Neat solution	TeA	y = 2477.96x + 3394.45	0.5–200			0.9973
	AOH	y = 1411.49x + 941.08	0.5–200			0.9959
	AME	y = 1817.16x + 322.29	0.5–200			0.9989
	TEN	y = 5284.07x−58.31	0.05–200			0.9969
	ALT	y = 524.42x + 638.12	1–200			0.9928
	ALS	y = 143.63x + 11.17	0.5–200			0.9982
	AFB_1_	y = 13387.40x + 2131.25	0.05–200			0.9978
	OTA	y = 12142.70x + 1612.83	0.1–200			0.9998
	ZEN	y = 2592.33x − 45.41	0.2–200			0.9956
Tomato	TeA	y = 3151.74x + 2661.33	0.5–200	0.20	0.50	0.9975
	AOH	y = 1381.90x + 88.99	0.5–200	0.20	0.50	0.9961
	AME	y = 1802.95x + 139.73	0.5–200	0.10	0.50	0.9968
	TEN	y = 5037.77x − 1339.19	0.05–200	0.01	0.05	0.9975
	ALT	y = 495.76x + 2523.89	1–200	0.30	1.00	0.9973
	ALS	y = 99.21x − 44.54	1–200	0.50	1.00	0.9933
	AFB_1_	y = 13910.30x − 3759.31	0.5–200	0.20	0.50	0.9957
	OTA	y = 11913.80x − 3011.72	0.1–200	0.05	0.10	0.9969
	ZEN	y = 1836.13x − 408.39	0.5–200	0.20	0.50	0.9966
Watermelon	TeA	y = 3147.83x + 3812.73	0.5–200	0.20	0.50	0.9988
	AOH	y = 1439.20x + 166.73	0.5–200	0.20	0.50	0.9998
	AME	y = 1942.00x + 660.55	0.5–200	0.10	0.50	0.9995
	TEN	y = 5411.94x−86.24	0.1–200	0.05	0.10	0.9995
	ALT	y = 463.38x + 2883.52	1–200	0.30	1.00	0.9952
	ALS	y = 147.57x − 105.27	1–200	0.50	1.00	0.9985
	AFB_1_	y = 14048.90x − 407.44	0.05–200	0.01	0.05	0.9998
	OTA	y = 12596.50x − 1507.82	0.5–200	0.20	0.50	0.9991
	ZEN	y = 2064.16x − 407.63	0.5–200	0.10	0.50	0.9923
Melon	TeA	y = 3590.11x + 3091.17	0.5–200	0.20	0.50	0.9981
	AOH	y = 1447.78x + 35.81	0.5–200	0.20	0.50	0.9973
	AME	y = 1832.77x + 632.65	0.5–200	0.10	0.50	0.9993
	TEN	y = 5228.12x−492.67	0.1–200	0.05	0.10	0.9993
	ALT	y = 446.22x + 6640.54	1–200	0.30	1.00	0.9976
	ALS	y = 110.85x − 74.61	1–200	0.50	1.00	0.9968
	AFB_1_	y = 13980.80x − 1061.58	0.1–200	0.02	0.10	0.9998
	OTA	y = 12393.30x − 462.63	0.5–200	0.10	0.50	0.9994
	ZEN	y = 2160.20x − 575.37	0.5–200	0.10	0.50	0.9931
Strawberry	TeA	y = 3323.10x + 3464.16	0.5–200	0.20	0.50	0.9972
	AOH	y = 1406.51x − 68.41	0.5–200	0.20	0.50	0.9983
	AME	y = 1869.42x + 594.67	0.5–200	0.10	0.50	0.9992
	TEN	y = 5245.70x − 460.86	0.1–200	0.05	0.10	0.9996
	ALT	y = 450.92x + 2765.14	1–200	0.30	1.00	0.9922
	ALS	y = 93.84x + 58.02	1–200	0.50	1.00	0.9940
	AFB_1_	y = 13843.00x − 1246.62	0.1–200	0.02	0.10	0.9996
	OTA	y = 11841.30x − 470.11	0.5–200	0.10	0.50	0.9993
	ZEN	y = 1792.03x − 54.64	0.5–200	0.10	0.50	0.9983
Hawthorn	TeA	y = 3292.18x + 5601.93	0.5–200	0.20	0.50	0.9958
	AOH	y = 1423.11x + 12363.80	0.5–200	0.20	0.50	0.9943
	AME	y = 1799.34x + 365.52	0.5–200	0.20	0.50	0.9996
	TEN	y = 5082.12x + 1587.85	0.05–200	0.02	0.05	0.9996
	ALT	y = 419.79x + 9686.70	1–200	0.30	1.00	0.9910
	ALS	y = 66.53x + 34.26	1–200	0.50	1.00	0.9918
	AFB_1_	y = 13483.20x − 304.85	0.1–200	0.02	0.10	0.9992
	OTA	y = 12097.60x − 1068.53	0.5–200	0.10	0.50	0.9993
	ZEN	y = 1426.21x − 169.96	0.5–200	0.10	0.50	0.9972

^a^ Limit of detection (S/N = 3). ^b^ Limit of quantification (S/N = 10).

**Table 2 toxins-15-00117-t002:** Spike recoveries, intra-day and inter-day precision of 9 mycotoxins in different matrices based on MSPE and HPLC-MS/MS methods.

Mycotoxins	Spiked Levels (μg kg^−1^)	Tomato			Watermelon			Melon			Strawberry			Hawthorn		
		Recovery (X ± SD, %) *n* = 3	Intra-Day Precision (Intra-RSD, %) *n* = 5	Inter-Day Precision (Inter-RSD, %) *n* = 5	Recovery (X ± SD, %) *n* = 3	Intra-Day Precision (Intra-RSD, %) *n* = 5	Inter-Day Precision (Inter-RSD, %) *n* = 5	Recovery (X ± SD, %) *n* = 3	Intra-Day Precision (Intra-RSD, %) *n* = 5	Inter-Day Precision (Inter-RSD, %) *n* = 5	Recovery (X ± SD, %) *n* = 3	Intra-Day Precision (Intra-RSD, %) *n* = 5	Inter-Day Precision (Inter-RSD, %) *n* = 5	Recovery (X ± SD, %) *n* = 3	Intra-Day Precision (Intra-RSD, %) *n* = 5	Inter-Day Precision (Inter-RSD, %) *n* = 5
TeA	2	95.8 ± 0.2	2.4	8.7	84.4 ± 0.2	4.9	9.2	94.6 ± 0.2	6.9	8.2	99.2 ± 0.2	5.0	10.6	96.3 ± 0.2	6.6	11.4
	50	90.8 ± 1.5	3.1	6.2	79.2 ± 0.4	5.4	3.7	84.5 ± 1.4	7.5	11.1	78.5 ± 1.4	4.2	11.7	96.4 ± 2.0	3.8	8.8
	100	85.6 ± 3.1	2.6	6.1	74.8 ± 3.7	4.9	8.6	81.1 ± 3.8	5.4	7.8	80.4 ± 5.7	7.6	10.7	88.7 ± 6.2	3.9	11.7
AOH	2	82.0 ± 0.1	3.8	4.3	88.3 ± 0.1	4.9	6.0	93.1 ± 0.2	3.8	12.4	82.0 ± 0.1	3.2	3.7	103.6 ± 0.1	2.1	9.64
	50	84.5 ± 1.3	4.3	12.9	92.4 ± 2.2	5.4	9.7	100.9 ± 4.5	3.4	7.4	101.9 ± 4.3	5.2	8.6	95.9 ± 4.3	6.9	8.7
	100	111.7 ± 3.4	3.8	5.1	94.2 ± 2.3	4.2	4.0	82.5 ± 4.9	7.7	4.6	83.2 ± 4.6	4.8	9.2	76.3 ± 4.5	4.3	9.7
AME	2	98.1 ± 0.2	3.1	7.5	89.1 ± 0.2	2.3	8.4	104.4 ± 0.2	3.9	5.3	87.7 ± 0.1	3.4	7.0	109.9 ± 0.2	6.2	5.81
	50	82.5 ± 1.1	2.8	8.4	78.4 ± 1.3	3.0	5.7	91.2 ± 2.6	4.3	9.6	96.5 ± 1.6	4.5	10.8	107.9 ± 1.3	5.2	8.3
	100	78.2 ± 3.9	5.4	8.4	81.5 ± 4.8	3.0	4.7	88.8 ± 2.9	4.1	5.4	91.7 ± 5.6	3.8	10.1	98.9 ± 3.1	5.4	5.2
TEN	2	89.7 ± 0.1	4.4	7.9	75.2 ± 0.1	5.7	2.2	82.8 ± 0.1	5.2	5.8	92.9 ± 0.1	7.2	11.9	88.1 ± 0.1	6.7	6.7
	50	91.5 ± 0.7	5.2	5.1	83.7 ± 1.5	6.4	11.7	96.6 ± 1.6	5.5	6.2	91.9 ± 1.3	6.7	9.3	92.0 ± 1.3	5.6	9.7
	100	75.4 ± 1.8	7.3	3.9	90.1 ± 2.3	6.2	4.3	75.7 ± 4.3	3.9	9.4	79.8 ± 4.5	5.8	9.5	82.5 ± 2.3	6.1	4.7
ALT	2	86.1 ± 0.1	7.9	7.2	80.4 ± 0.2	8.6	7.6	81.1 ± 0.1	3.3	9.1	99.9 ± 0.2	4.0	12.7	96.8 ± 0.2	4.6	10.7
	50	90.8 ± 1.4	6.2	4.4	81.5 ± 1.4	7.5	11.8	75.6 ± 1.1	5.5	9.3	78.5 ± 1.4	7.2	11.7	99.4 ± 2.0	5.3	9.3
	100	85.6 ± 3.1	6.1	11.5	76.9 ± 2.7	7.5	6.2	75.6 ± 2.4	6.8	5.2	74.9 ± 1.7	8.1	3.8	77.6 ± 2.2	4.6	4.8
ALS	2	93.0 ± 0.1	3.6	5.9	85.7 ± 0.2	4.3	3.0	87.8 ± 0.1	2.7	4.2	83.6 ± 0.2	8.6	6.6	84.7 ± 0.1	4.8	11.0
	50	86.8 ± 1.9	4.1	6.8	90.3 ± 1.3	2.2	4.2	88.2 ± 1.7	7.5	10.4	88.0 ± 1.8	3.0	11.4	88.4 ± 1.3	5.1	8.9
	100	88.71 ± 5.84	4.7	6.1	83.1 ± 4.0	5.4	6.1	91.3 ± 5.3	5.7	9.1	90.7 ± 4.9	3.8	8.9	89.9 ± 5.6	4.4	4.2
AFB_1_	2	81.1 ± 0.1	2.9	8.4	81.3 ± 0.1	7.9	10.6	96.6 ± 0.1	6.1	5.1	102.1 ± 0.2	7.1	8.7	97.1 ± 0.1	3.6	8.4
	50	83.6 ± 0.9	3.9	7.6	79.0 ± 1.8	4.4	8.9	79.6 ± 1.9	8.6	9.5	85.4 ± 1.7	5.9	10.9	85.3 ± 1.7	5.8	8.9
	100	77.2 ± 2.6	4.6	6.1	82.5 ± 5.9	5.4	11.5	74.3 ± 4.7	5.5	8.9	80.6 ± 2.9	6.3	5.9	77.4 ± 5.1	5.0	10.9
OTA	2	108.1 ± 0.1	6.2	5.4	100.6 ± 0.1	4.5	3.5	89.8 ± 0.2	6.7	12.3	75.6 ± 0.1	5.5	8.5	90.6 ± 0.2	3.3	9.2
	50	111.7 ± 1.3	5.7	7.5	97.3 ± 1.6	7.4	11.0	99.6 ± 1.8	4.2	11.9	93.6 ± 1.7	3.8	11.80	98.5 ± 1.6	6.9	11.0
	100	84.1 ± 4.4	6.0	8.7	76.3 ± 5.6	6.9	12.3	75.7 ± 1.9	7.7	4.2	83.1 ± 2.5	2.1	4.9	83.8 ± 1.9	7.8	3.8
ZEN	2	78.8 ± 0.2	4.3	9.6	91.2 ± 0.1	5.2	5.0	107.5 ± 0.1	3.8	4.6	84.8 ± 0.2	7.2	9.9	78.2 ± 0.4	6.2	3.2
	50	99.7 ± 1.5	7.1	10.0	84.8 ± 1.6	9.0	8.4	101.7 ± 1.5	4.2	9.9	76.6 ± 1.8	6.7	11.2	79.2 ± 0.7	5.0	5.8
	100	85.3 ± 2.8	6.2	5.4	91.2 ± 4.0	4.8	7.4	87.1 ± 2.3	3.8	4.4	82.0 ± 4.9	5.7	3.8	87.1 ± 3.3	5.9	6.4

## Data Availability

Not applicable.

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
