# Peer review of "Fe3O4@COF(TAPT–DHTA) Nanocomposites as Magnetic Solid-Phase Extraction Adsorbents for Simultaneous Determination of 9 Mycotoxins in Fruits by UHPLC–MS/MS"

_toxins, 2023, doi:10.3390/toxins15020117_

Round 1

Reviewer 1 Report (Previous Reviewer 1)

After reviewing the author’s response and justifications, I believe this study is suitable to publish in Toxins, however, this method could be useful for me research only.

Author Response

Thank you for your useful suggestions. 

Reviewer 2 Report (Previous Reviewer 3)

This paper has improved compared to the previous version.  Still, it has some issued that need to be addressed.

Statistical Analysis.  "Significant tests were conducted by the analysis of variance (ANOVA) followed by Duncan’s least significant test.”  Well, before using ANOVA the authors need to check for normality and homoscedasticity.  It is not big deal, and you can do it.  

The sentence containing "the trueness of data" should be re-written using "assessment of method accuracy".

Please separate the () from "detection": Limit of detection(S/N=3)

What are x and y in the formulas in Table 1?  For example, y=2477.96x+3394.45.

Figure 3 F. Change "Concentration of NaCl (mg/l)" to "NaCl concentration (mg/L)".

Figure 3A. Please separate the () from the words.  Check Figure 2 on the same issue [2C, x-axis legend is cut; 2F, check the x-axis legend].  Some words are not completely visible.

The scale in Fig 2 A,B is not clear.  Please take your time and check carefully the quality of all Figures.  That is important. 

Check grammar in Figure 1 legend.

Author Response

Reviewer 3 Report (New Reviewer)

The research is original and interesting, and the manuscript is well-organized and nicely written. However, there are only very minor revisions to be made (see attached file) in this manuscript before the publication in Toxins.

Author Response

Reviewer 4 Report (New Reviewer)

In the submitted manuscript, the authors developed a new method for monitoring mycotoxins in fruits and berries. The technique is substantiated, worked out, the conditions are optimized. The experimental model was tested on several matrices and several mycotoxins. In my opinion, the article is very interesting and should be published. However, the data obtained are not discussed in any way. After the presentation of the results, conclusions are immediately drawn, which makes the article seem incomplete.

Author Response

Thank you for your useful suggestions. I'm very sorry for the misunderstanding. The results and discussion had been presented in section of “2. Results” in original manuscript. According to your suggestion, the description of “2. Results” has been changed to “2. Results and discussion” in the revised manuscript.

This manuscript is a resubmission of an earlier submission. The following is a list of the peer review reports and author responses from that submission.

Round 1

Reviewer 1 Report

This MS aims to develop a method (extraction method MSPE) for determination of some mycotoxins in fruit using MS system. Compared with other available methods such as QuEChERS and other immunoaffinity column SPE, this considered as a very long and complicated process for sample preparation -120 min or more. Also, there are 2 steps of evaporation and re-constitution that increase the chance of losing the analytes and overall, the in-lab applicability would be very limited. We are shooting for “dilute and shoot” for determination and detection of food contaminants such as mycotoxins and looking for cheap, fast and easy method for extraction and sample preparation (< 10 min). I recommend the authors to compare the proposed method with other reference methods such as AOAC. Also. I recommend running some CRM samples with the collaboration of other labs. My big concern is the applicability of this method and the usability since the preparation time would be around 160 min.

The resolution of all graphs is very low

Reviewer 2 Report

The proposed article addresses a topic of interest, however, for its publication in the journal it is necessary to take into account the following recommendations:

1.       You should consider the length of the topic proposed for this research article. The topic is very broad and intends to cover everything done within the research. Decrease the number of words for it, it is not necessary to be so specific.

1.       Be cautious in the abstract when mentioning recovery percentages. Make it clear that these values correspond to the recovery percentage of several analytes and not only to one analyte after elution.

2.       It would be interesting to describe the adsorption mechanism of adsorbates in a general way including in a figure where π-π type interactions are represented. It could be supported by FTIR characterization of the material before and after the adsorption process. The vibrations and stretching of the bands corresponding to the aromatic rings can give a good idea of corresponding to this mechanism.

3.       Include for this work a cycling study that considers the reuse of the material after the first extraction. It is essential to have an idea of the number of cycles that the material can be used without losing its effectiveness.

Reviewer 3 Report

The authors of this paper prepared and used magnetic solid-phase extraction (MSPE) to purify and enrich  mycotoxins present in fruits using a nanomaterial.  The material, Fe3O4@COF(TAPT-DHTA), was able to capture targeted mycotoxins, and different conditions of the extraction were revised and optimized. The method was validated employing different fruit samples following mycotoxin analysis by UHPLC-MS/MS.

The paper is interesting, but some points need attention, in particular the lack of statistics.

What is the average concentration of mycotoxins in fruits based on literature data?  Is it 20 µg/kg?  If not, why using this concentration.

I am really worried because the authors choose the final extraction parameters based solely on the Figure (3), using the “eye”, without considering any statistical support.  This is the main reason I am using to reject the paper.

Another major concern in this study is the design of the validation protocol. It seems all variables were check individually, lacking proper interaction between them. Please address this problem.  

Moreover, the statistics for the data on Figure 3 should be shown. 

Figure 1 should be improved.  Words are not clear.

Figure 2 is not good. It needs to be improved. Numbers are difficult to read, in particular at 2F.

Figure 3 is also difficult to read.

Table 1. Linear range vs. Linear annge.

Table 2. Report only one digit.
